# Swim Training Ameliorates Hyperlocomotion of ALS Mice and Increases Glutathione Peroxidase Activity in the Spinal Cord

**DOI:** 10.3390/ijms222111614

**Published:** 2021-10-27

**Authors:** Katarzyna Patrycja Dzik, Damian Józef Flis, Zofia Kinga Bytowska, Mateusz Jakub Karnia, Wieslaw Ziolkowski, Jan Jacek Kaczor

**Affiliations:** 1Department of Animal and Human Physiology, Faculty of Biology, University of Gdansk, 80-309 Gdansk, Poland; katarzyna.dzik@ug.edu.pl (K.P.D.); mateusz.karnia@ug.edu.pl (M.J.K.); 2Department of Pharmaceutical Pathophysiology, Faculty of Pharmacy, Medical University of Gdansk, 80-211 Gdansk, Poland; damian.flis@gumed.edu.pl; 3Division of Bioenergetics and Physiology of Exercise, Faculty of Health Sciences with Institute of Maritime and Tropical Medicine, Medical University of Gdansk, 80-211 Gdansk, Poland; zofia.bytowska@gumed.edu.pl; 4Department of Rehabilitation Medicine, Faculty of Health Sciences with Institute of Maritime and Tropical Medicine, Medical University of Gdansk, 80-219 Gdansk, Poland; wieslaw.ziolkowski@gumed.edu.pl

**Keywords:** hyperlocomotion, ALS, oxidative stress, metabolism, spinal cord

## Abstract

(1) Background: Amyotrophic lateral sclerosis (ALS) is an incurable, neurodegenerative disease. In some cases, ALS causes behavioral disturbances and cognitive dysfunction. Swimming has revealed a neuroprotective influence on the motor neurons in ALS. (2) Methods: In the present study, a SOD1-G93A mice model of ALS were used, with wild-type B6SJL mice as controls. ALS mice were analyzed before ALS onset (10th week of life), at ALS 1 onset (first symptoms of the disease, ALS 1 onset, and ALS 1 onset SWIM), and at terminal ALS (last stage of the disease, ALS TER, and ALS TER SWIM), and compared with wild-type mice. Swim training was applied 5 times per week for 30 min. All mice underwent behavioral tests. The spinal cord was analyzed for the enzyme activities and oxidative stress markers. (3) Results: Pre-symptomatic ALS mice showed increased locomotor activity versus control mice; the swim training reduced these symptoms. The metabolic changes in the spinal cord were present at the pre-symptomatic stage of the disease with a shift towards glycolytic processes at the terminal stage of ALS. Swim training caused an adaptation, resulting in higher glutathione peroxidase (GPx) and protection against oxidative stress. (4) Conclusion: Therapeutic aquatic activity might slow down the progression of ALS.

## 1. Introduction

Amyotrophic lateral sclerosis (ALS) is the most frequent chronic motoneuron disease, characterized by progressive motor weakness, muscle atrophy, and selective motoneuron loss. About 90–95% of ALS cases have an unknown origin, and this form of ALS is called sporadic (SALS). ALS may also be determined genetically as the familial form of ALS (FALS). FALS has the same clinical symptoms as SALS, although its development starts at a younger age [1]. In 1993, Rosen showed the connection between some cases of FALS and the mutations in superoxide dismutase 1 (SOD1) with the locus on the 21st chromosome. SOD1 mutation has been observed in about 10% of FALS patients. SOD1 is an antioxidant enzyme protecting not only neurons but the whole organism from superoxide ions, which in the mutated form may cause oxidative stress and stimulate protein aggregation leading to cell death [2]. Besides antioxidative function, SOD1 protects cells by regulating cell energy and metabolism, therefore its disturbances affect the homeostasis of the whole organism [3]. SOD1-G93A mice, with mutant human SOD1 cDNA inserted randomly into the mouse genome, are one of several mice models used in ALS studies. Apart from oxidative stress [4], reduced basal mitochondrial respiration and the concomitant elevation of glycolysis have been shown as metabolic hallmarks of ALS motor neurons [5]. Decreasing oxidative stress [4,6] and improving mitochondrial function [5,7] seem to be the most important strategies for treating ALS.

A recent meta-analysis suggests that exercise, especially endurance/aerobic exercise, can be a beneficial and safe option for patients with ALS. However, the condition to recommend a specific type of exercise for ALS patients is the low cost of the exercise [8]. A study comparing the effects of running and swimming in SOD1-G93A mice has shown that the neuroprotective impact of swimming on motor neurons outranked running. Swimming has been shown to increase lifespan [7,9], reduce neuromuscular junction degeneration and brain-derived neurotrophic factor degeneration, as well as to attenuate the drastic loss of big and fast motoneuron in SOD1-G93A mice [10]. Following the benefits of low-impact exercise, it was found that voluntary physical activity, motivated by the change of the environment, induces similar benefits to those of swimming, probably with lowering oxidative stress compared with forced running protocols [11,12].

Alterations in most neurotransmitter systems can give rise to a hyperactive phenotype, representing hyperlocomotor activity [13]. In ALS, apart from hypermetabolism [14,15,16,17,18,19], the increased locomotor activity has been shown in SOD1-G93A mice at the pre-symptomatic stage of the disease [20]. Moreover, a study conducted in northern England between 2009–2013 on 175 patients with ALS and 350 healthy controls showed that patients with ALS undertook higher levels of physical activity than controls, both work-related and in their spare time. The authors of this study documented that participation in additional physical activities was equal to 10 kJ/kg/day (this is about 45 min of fast walking), which was consistently associated with an increased ALS risk [21].

Therefore, the present study investigates the behavioral changes associated with ALS disease progression and the modifications of these changes induced by swim training intervention. Additionally, our goal is to explore the concentration of oxidative stress markers and antioxidant and energy metabolism enzyme activities in the spinal cord of SOD1-G93A mice.

## 2. Results

### 2.1. The Suppressing Effect of Swim Training on Hyperlocomotion in ALS Mice 

Our team had previously shown that swim training undoubtedly increases handgrip strength in ALS mice in the early symptomatic stage of the disease [22]. Moreover, we have shown that swim training increased survival in ALS mice by 10 days, from 126 days of the lifespan in control ALS mice to 135.5 days of the lifespan of ALS trained mice [7]. Here we question whether swim training modifies the behavior of ALS mice. To address this matter, we assessed all mice in the open field test. This examination revealed over an 87% greater number of lines were crossed by mice from the ALS control group when compared to the WT before group, during the first exposure to an open field test environment. This result indicates basal hyperlocomotion of ALS mice. Furthermore, the ALS mice, after three weeks of swim training, crossed significantly lower distances during the first exposure to the open field test as compared to the ALS control groups (both ALS 1 onset and ALS TER) (*p* < 0.05). The ALS SWIM groups (both ALS 1 onset SWIM and ALS TER SWIM) crossed an average of 73.6 ± 14.7 lines, which is almost 4-fold less than ALS control groups, which crossed 292.0 ± 63.7 lines (Figure 1A). Moreover, a reduction of the number of lines crossed after swim training was observed during four subsequent weeks (Figure 1B). Interestingly, in the WT SWIM group (both WT 1 onset SWIM and WT TER SWIM) we did not find any impact of swim training either during the first observation or the following four observations. It is important to note that in the WT mice, we did not observe any significant impact of swim training during the first open field test nor the subsequent test. Nevertheless, we observed overall decreased activity in all mice in the repeated exposure to the open field test sessions, probably due to the loss of the novelty of the test environment [23].

### 2.2. The Effect of Swim Training and Disease Progression on the Energy Metabolism of the Spinal Cord in ALS Mice

We have previously shown several significant modulations in the metabolism of skeletal muscle of ALS mice concerning the application of swim training. Here we investigated the impact of swim training on ALS-induced changes in the spinal cord to achieve comparative information for skeletal muscle. In addition to the behavioral hyperactivity, we have detected higher citrate synthase (CS) activity in pre-symptomatic ASL mice (*p* < 0.05) (Figure 2B), indicating the improvement of oxidative energy metabolism when compared to WT mice. With the progression of the disease, the activity of CS was consecutively declining to a final activity of 40% lower at the terminal stage of the disease in the ALS TER group. However, the swim training had ameliorated these changes as the ALS TER SWIM group showed only a 28% decline of CS activity when compared to the ALS before group.

Additionally, lactate dehydrogenase (LDH) activity in pre-symptomatic mice in the ALS before group was higher than in the WT before group. Correspondingly to CS activity, we have observed a decline of LDH activity at the first onset of the disease, but only in the ALS 1 onset group. However, at the terminal stage of the disease in the spinal cord LDH activity was increased, which suggests that the energy demands in nervous tissue with the shortage of ATP from oxidative energy sources may be fulfilled from glycolytic sources. In the swim training groups, we found higher LDH activity both in the ALS 1 onset SWIM and the ALS TER SWIM groups compared to the corresponding untrained groups (Figure 2A). The LDH activity in the ALS TER SWIM group averaged 153.10 ± 23.34, which was over twice as much as in the WT before group (69.46 ± 19.39) and almost twice as much as in the ALS before group (87.33 ± 20.63 nmol/min/mg protein). These changes indicate a strong influence of swim training towards meeting the energy demands of nervous tissue at the terminal stage of ALS.

We have recently shown that in the tight skeletal muscle, both CS and LDH activities were lower in ALS pre-symptomatic groups than the WT corresponding groups. Therefore, we assume that the hyperlocomotion of ALS mice has sequelae mainly on nervous tissue. No difference was found between WT groups in CS and LDH activities.

### 2.3. The Modified Enzymatic Antioxidant System in the Spinal Cords of ALS Mice

To determine whether the enzymatic antioxidant defense changes upon ALS progression and swim training, we assayed the activity of major antioxidant enzymes: total superoxide peroxidase (SOD), SOD1, superoxide peroxidase 2 (SOD2), catalase (CAT), and GPx. The total SOD activity in the spinal cord of ALS pre-symptomatic mice was 5.9 times higher than in the WT group (*p* < 0.005) (Figure 3A). In the SOD1 activity, this disproportion was even more pronounced since the ALS mice SOD1 activity was 6.3-fold higher than in the WT group (*p* < 0.005) (Figure 3B); whereas the SOD2 activity in the ALS before group was 4.5 times higher than in the WT group (*p* < 0.005) (Figure 3C). With the disease progression, SOD1 activity significantly increased both at the first onset of the disease stage and the terminal stage (*p* < 0.05). Swim training initially increased the activity of SOD1. In the ALS 1 onset SWIM group, the SOD1 activity was 13% higher than in the ALS 1 onset control group. However, at the terminal stage SOD1 activity in the trained mice was slightly lower than in the ALS TER control group. In the SOD2 activity, we did not find any alterations in the progression of the disease. Despite that, we observed a slightly higher SOD2 activity in the ALS 1 onset SWIM group.

The GPx activity in pre-symptomatic ALS mice spinal cords was significantly lower than in WT mice, and it was decreased by 2-fold in comparison to the healthy WT mice (*p* < 0.05) (Figure 4A). Given the over 6-fold higher SOD1 activity, these results show that GPx activity might not be required for maintaining oxidative balance at the pre-symptomatic stage of ALS. However, with disease progression, at the first onset of the disease, we found 2.6-fold higher GPx activity, 1786.0 ± 78.42 in the ALS before group, and 4703.0 ± 268.8 nmol/min/mg protein in the ALS 1 onset group (*p* < 0.05). At the terminal stage of the disease, the GPx activity dropped again to the initial level (1943.0 ± 69.6 nmol/min/mg protein). However, the swim training brings a significant effect on GPx activity. At the first onset of diseases, the trained group already showed elevated GPx activity compared to the ALS 1 onset group. However, the most remarkable change was observed at the terminal stage of ALS. In the ALS TER SWIM group, 115% higher GPx activity was found than in the ALS TER group (*p* < 0.005) (Figure 4A). 

Next, we investigated if the GPx activity was dependent on the reduced glutathione (GSH) availability. Despite low GPx activity at the pre-symptomatic stage of the disease, the GSH concentration in the spinal cord in the ALS before group was only slightly lower than in the WT before group, and it was respectively 8.62 ± 0.2 in the ALS before group and 9.4 ± 0.5 μM in the WT before group.

Nevertheless, with the disease progression, the GSH concentration gradually declined to the 7.8 ± 0.4 μM at the terminal stage in the ALS TER group (*p* < 0.05). The swim training seems to stop the lowering of the concentration GSH. However, the observed trend was not statistically significant (Figure 4B). 

The activity of CAT in the WT before group was lower than in all ALS groups. The CAT activity was not changed at the first onset of the disease. However, at the terminal stage of the disease, the CAT activity significantly rose (*p* < 0.05) (Figure 4C). The swim training seems to increase the activity of CAT activity at the first onset of the ALS disease, yet at the terminal stage, it had the opposite effect. The CAT activity in the ALS TER SWIM group was lower than in the ALS TER group. It is important to note that elevated CAT activity in the ALS TER group was associated with lower GPx activity and lower GSH concentration. No difference was found between WT groups in GPx and CAT activities and the concentration of GSH.

### 2.4. The Attenuation of Free Radical Damage in the Spinal Cord of ALS Mice after Swim Training

The range of free radical damage was examined with three markers: the sulfhydryl groups (the SH groups), 8-Isoprostanes, and 8-OH guanosine. The WT mice group showed a higher concentration of the SH groups than the pre-symptomatic ALS group. At the first onset of disease the SH groups concentration decreased, indicating higher protein damage in the spinal cord. The SH groups concentration in the ALS 1 onset group was significantly lower than in the WT group (*p* < 0.05) (Figure 5A). However, at the terminal stage of ALS, we observed lower protein damage, pronounced with the higher SH groups concentration in the ALS TER group versus the ALS 1 onset group. This may indicate that the protein damage had already reached a steady state due to the degradation of heavily oxidized proteins in earlier disease stages. Our finding is consistent with the previous reports showing that the spinal cords of G93A rats exhibited higher total concentrations of carbonylated proteins than wild-type rats. However, the concentrations did not significantly increase from the pre-symptomatic to the symptomatic phase [24]. The swim training escalated the protein damage but only at the first onset of the disease. After swim training, the terminal ALS group showed higher SH group concentration, which may indicate that swimming either caused the steady state of protein degradation earlier in the ALS TER SWIM group or a protecting mechanism evolved. With the significantly higher GPx activity in the ALS TER SWIM group, as compared to the ALS TER group, we may assume that GPx plays a protecting role against protein degradation in the spinal cord of ALS mice.

The concentration of the SH groups negatively correlated with the concentration of 8-Isoprostanes (r = −0.35, *p* < 0.05) (Figure 5C). We found a similar direction in lipids peroxidation as was described above in proteins. With the disease progression, the concentration of 8-Isoprostanes increased. In the ALS TER group, it was 45% higher than in the ALS before group (*p* < 0.05) (Figure 5B). The swim training withdrew the aggravation of lipid peroxidation since we observed a lower concentration of 8-Isoprostanes in the trained groups. The preventive influence of swim training was more pronounced in the ALS TER SWIM group.

The concentration of 8-Isoprostanes correlated with the concentration of 8-OH guanosine (r = 0.447, *p* < 0.05) (Figure 5E). The concentration of 8-OH guanosine in the ALS TER group was significantly higher than in the ALS 1 onset group. The swim training seems to have a protective effect against DNA damage since in the ALS TER SWIM group the 8-OH guanosine concentration was lower than in the ALS TER group (Figure 5D). No difference was found between WT groups in the concentration of oxidative stress markers.

## 3. Discussion

The main finding of the present study is that the swim training-induced reduction of locomotor hyperactivity occurred concomitantly with metabolic changes in the spinal cord. The swim training ameliorated the oxidative damage in the spinal cord and changed the antioxidant enzyme activities. The most profound characteristic of the spinal cord in ALS mice was the large increase of SOD’s isoenzymes activities in all of the stages of the disease. The higher CAT and reduced GPx activities at the terminal stage ALS were also observed. Besides this, it was found that swim training augmented GPx activity, especially in the terminal stage of ALS, suggesting that GPx in the spinal cord works physiologically and can adapt to the exercise stimulus.

Hypermetabolism is a hallmark of many patients with ALS. Several reports have demonstrated that patients with ALS have increased resting energy expenditures [14,15,16,17,18,19]. Interestingly, for FALS patients, the probability of being hypermetabolic is higher than for SALS patients. The study of Bouteloup et al. [14] showed that 11/11 patients with FALS were hypermetabolic, compared to 17/33 patients with SALS, and the resting energy expenditure was significantly higher in FALS than in SALS patients. Although hypermetabolism is the most often analyzed for energy expenditure and weight loss, it has been recently associated with greater functional decline and shorter survival in patients [18] as well as cognitive impairment in ALS and frontotemporal dementia (FTD) [25,26]. Moreover, the changes in metabolic status can be determined early on in the stages of ALS [14]. Many new ALS models are emerging to represent different mutations, some of them show major behavioral disturbances. The study on SOD1-G93A mice showed that pre-symptomatic ALS mice exhibited a propensity to run long distances and then undergo a precipitous fall in the daily running distance before the onset of observable motor impairment. Importantly, in that study, the WT mice traveled almost three times shorter daily distances than ALS pre-symptomatic mice. However, at the first onset of the disease, the daily running distance of WT mice exceeded the daily running distance of ALS mice [20]. Additionally, TLS-/-knockout mice showed locomotor hyperactivity and reduced anxiety-related behavior, both of which were supported by multiple different tests [27]. In the present study, we found that ALS pre-symptomatic mice had increased locomotor activity when compared to the healthy WT mice. Furthermore, we observed that swimming already remarkably reduced locomotor activity in ALS mice after three weeks of training (Figure 1A). Our previously published data clearly stated that swim training positively impacted lifespan [7] and handgrip of ALS mice [22]. The same training protocol, previously described by Deforges and coworkers [9], was used to compare the efficiency of swimming- and running-based training in SOD1-G93A mice. Although both types of exercise had induced neuroprotective mechanisms, unlike running, swimming significantly delayed spinal motoneuron death and the motoneurons of large soma areas. They also observed that the swimming-based training specifically promoted significant maintenance of the oligodendrocyte population, which displayed a 30% loss in sedentary and running mice. This observation was recently confirmed by another study using the same training procedure [10]. Additionally, it was stated that swimming might reduce neuromuscular junction degeneration and brain-derived neurotrophic factor degeneration and soften the drastic loss of big and fast motoneurons in SOD1-G93A mice. The possible explanation to swimming’s neuroprotective impact might be the fact that swimming-based training, which is a high-amplitude and frequency exercise, recruits fast motor units integrated by large motoneuron, whereas running-based training, which is a low-amplitude and frequency exercise, preferentially triggers slow motor units integrated by small motoneurons [9,28]. Swimming exclusively preserves big motoneurons, with a soma area between 900 and 1200 μm^2^, which are more susceptible to oxidative stress and excitotoxicity induced by physical activity [29]. Indeed, running is a high-impact exercise that only recruits small motoneurons and generates more oxidative stress than swimming, a low-impact exercise that recruits both small and big motoneurons. Therefore, it is comprehensible that, despite both protocols inducing molecular changes at the motoneuron terminal, swimming prevented motoneuron loss while running did not [9,10]. Moreover, swimming exercise has been shown to positively influence behavioral outcomes in various cases. For instance, in an animal model of ADHD, swimming caused the suppression of hyperactivity, impulsivity, and non-aggressive and aggressive behaviors, and alleviation of the short-term memory impairment in rats [30], as well as the improvement of mental health parameters, cognition, and motor coordination in children [31]. Recently, swim training has been shown to reduce the expression of A2A receptors [32], which overexpression resulted in increased total distance traveled and time spent in the center during an open field test [33]. The study on SOD1-G93A mice showed an exacerbation of A2A receptor-mediated excitatory effects at the pre-symptomatic phase, whereas, in the symptomatic phase, A2A receptor activation was absent [34]. Therefore, a swim beneficial training effect for pre-symptomatic ALS mice might involve the reduction of A2A receptors.

Apart from the hyperlocomotor activity in the ALS pre-symptomatic mice, we detected higher CS and LDH activity as compared to the WT before group. This seems to be an adaptation to the increased activity of the nervous system in the pre-symptomatic stage. A similar observation was made regarding CS activity in motor neurons isolated from pre-symptomatic SOD1-G93A mice where increased expression of genes involved in mitochondrial machinery, including ATP synthase, was found [35]. This could be due to the early compensatory activation of energy-generating pathways to counteract increased energy demand [36]. CS activity decreases with the ALS progression which the disruption of mitochondrial functions may partly explain. Tricarboxylic acid (TCA) cycle enzymes are inhibited during increased cellular oxidative stress as a possible adaptive response against reactive oxygen species (ROS) generation and to provide antioxidant effects. The inhibition of TCA cycle enzymes results in lower NADH generation, and diminished availability of electrons to enter the electron transport chain [37], and overproduction of superoxide anion. On the other hand, ROS are important signaling molecules, which may induce the adaptive response to the exercise. The increase of CS activity in the ALS TER SWIM group might suggest, that mitochondria are the potential source of ROS inducing a further adaptive response.

The LDH activity after dropping at the first onset of the disease increases its activity at the terminal stage of ALS. A similar observation was previously described by Miyazaki et al. [38], who showed that in the spinal cord of SOD1-G93A mice, glucose metabolism appeared to be reduced at early symptomatic and end stages of the disease but was increased before disease onset. Recently, another study used simultaneous [^18^F]-FDG PET magnetic resonance imaging and showed that glucose metabolism is decreased in the motor and somatosensory cortices of TDP-43 (A315T) mice, while at the same time, it was increased in the midbrain region between at pre-symptomatic age, as compared to WT controls [39].

Swim training applied in the present study increased the CS and LDH activity both at the first onset and at the terminal stage of ALS. Nevertheless, the LDH activity in both groups at the terminal stage was higher than in the ALS before group, but in the ALS TER SWIM group it almost doubly surpassed the LDH activity in the ALS before group. Simultaneously, the CS activity in the ALS TER SWIM group was lower than in the ALS before group. This clearly shows that in ALS, impaired oxidative metabolism is compensated by glycolysis. Among the common perturbations of ALS are a reduction in glucose uptake, impairments in glycolysis, and disrupted mitochondrial function, as well as ATP production. Compounds that activate glucose metabolism, mainly via glycolysis, may potentially slow disease progression [36]. This seems to be a protective influence of swim training adaptation to the increased energy demand.

The activity of GPx, which was doubly increased in the ALS TER SWIM group compared to the untrained ALS TER group, is another positive adaptation to swim training. This response might be possible due to maintained mitochondrial function, and ROS production, possibly at the physiological level. The lower GPx activity, when compared to healthy controls, was observed in the plasma of ALS patients and was associated with a higher disease progression rate [40]. Additionally, in postmortem brain homogenates, the 40% reduction in GPx activity was found to be potentially damaging and implicated in the pathogenesis of sporadic ALS [41]. Therefore, the post-training increase of GPx in this study seemed to be a physiological antioxidative defense mechanism (Figure 4A). However, the CAT activity showed no adaptation to the training. On the contrary, CAT activity was increased in the ALS TER group (Figure 4C), which occurred in response to elevated hydrogen peroxide (H_2_O_2_) concentration. The lack of consistency in the activity of both enzymes might be a consequence of different Km values for H_2_O_2_. CAT has a much higher affinity with H_2_O_2_ at greater concentrations than GPx [42], making GPx more sensitive to low-grade changes while CAT works at the pathologically high H_2_O_2_ concentration. Those results correspond with a higher concentration of oxidative stress markers in the ALS TER group (Figure 5A–C). The reduction of oxidative stress markers, particularly 8-Isoprostanes, in the ALS TER SWIM group might be a result of increased GPx activity at the same time point. The presence of oxidative stress in ALS has been repeatedly reported. Our results from skeletal muscles of ALS mice have shown that the damage of macromolecules in skeletal muscle intensifies with disease progression, yet swim training significantly reduces those changes [7]. The present study shows a similar tendency in the spinal cord of ALS mice. Interestingly, the increase of oxidative stress does not seem to start in the pre-symptomatic stage of ALS but at the first onset of the disease (Figure 5A–C), which corresponds with the GPx activity that stays at the low level in the ALS before group, but escalates at the first onset of disease, despite the application of training procedure (Figure 4A). 

Lastly, a significant increase of SOD1 and SOD2 activity in the spinal cord of all ALS groups as compared to the WT groups was found. A large increase of SOD’s isoenzymes activity was found in skeletal muscle [43], and SOD2 activity was also elevated in the central nervous system [44]. The causative SOD1 gain of function in ALS pathogenesis is indisputable, and several mechanisms by which this occurs have been proposed and comprehensively reviewed [45,46,47]. Moreover, it has been postulated that misfolding and/or aggregation is the most likely source of SOD1 toxicity [48]. However, SOD1 was found to be highly resistant to inhibition of its activity even in the misfolded and/or aggregated state [49].

A major limitation of the study is a lack of available blood samples at various time points for the assessment of corticosterone concentration, as the end product of hypothalamic-pituitary-adrenal (HPA) axis, for determination of stress levels.

## 4. Materials and Methods

### 4.1. Animals

Transgenic mice (B6SJL-TgN[SOD1-G93A]1Gur), an animal model of ALS, and wild-type male B6SJL mice serving as controls for this mutant strain were purchased from Jackson Laboratory (Bar Harbor, ME, USA), where mice were examined by genotyping. Upon arrival, mice passed a two-week adaptation period, after which they were randomly assigned to ALS before, ALS 1 onset, ALS 1 onset SWIM, ALS TER, and ALS TER SWIM, WT before, WT 1 onset, WT 1 onset SWIM, WT TER, or WT TER SWIM (Figure 6).

### 4.2. Housing and Euthanasia 

The mice were housed in an environmentally controlled room (23 ± 1 °C with a 12 h light-dark cycle, humidity 55%); the mice received standard mouse chow and water ad libitum. The mice were euthanized by cervical dislocation. The mice from the ALS before and WT before groups were euthanized on the 70th day of life. The ALS 1 onset were euthanized when the first symptoms of disease were observed (116 ± 2 days of life). The mice from ALS 1 onset SWIM, WT 1 onset, and WT 1 onset SWIM groups were sacrificed at the same age as the ALS 1 onset group. The mice from the ALS TER group were euthanized at the terminal stage of the disease (i.e., functional paralysis in both hind legs). The ALS TER SWIM, WT TER, and WT TER SWIM mice were sacrificed at the same age as the ALS TER mice.

### 4.3. Swim Training Protocol

Starting at the age of 10 weeks, transgenic (group ALS 1 onset SWIM and ALS TER SWIM) and wild type control mice (group WT 1 onset SWIM and WT TER SWIM) underwent training procedure = 5 times a week, according to Deforges et al. [9], with a slight modification described by Flis et al. [7]. Briefly, the training procedure was: 30 min swim, five times per week, in the water at the temperature 30 °C, without additional weight, in a swimming pool with an adjustable flow (max. 5 L min^−1^).

### 4.4. Sample Preparation and Material Collection

All the harvested spinal cords were collected, quickly frozen on liquid nitrogen, and then stored at −80 °C. For experiments, weighted pieces of spinal cords were homogenized using a glass homogenizer (4% *w*/*v*) in lysis buffer (50 mM Tris-HCl, 150 mM NaCl, 1 mM EDTA, 0.5 mM DTT) containing EDTA-free Protease Inhibitor Cocktail (04693159001, Roche, Switzerland) and PhosSTOP^™^phosphatase inhibitors (04906837001, Roche, Switzerland). Homogenates were centrifuged at 750 *g,* and then part of the obtained supernatants was spun at 5000 *g* for 10 min at 4 °C. For the analysis of free radical damage, a BHT solution was added (1 μL of 0.5% BHT for 10 μg of tissue). The resulting supernatant was decanted and frozen for further analysis. All samples were coded, and the staff members remained blind for the group assessment for all of the measurements in the spinal cord. Protein concentration was measured with the Bradford method.

### 4.5. Open Field Test

Open field test sessions were performed for the assessment of general exploratory locomotion in a novel environment. Each behavioral task was performed during the light phase. Mice were placed in a transparent Plexiglas (50 × 50 × 20 cm) arena with a white floor divided into 25 equal squares of 10 cm edge. Mice were allowed to freely adapt to the open field environment for 2–3 min. The behavioral activity was recorded for 5 min with a computer-linked video camera mounted above the testing box. The number of the line crossings was scored manually by the experimenter, who remained blinded for the mice’s group assessment. A line crossing was counted only when the animal crossed the line with all four paws. Room illumination was kept at 50 lux. The area used in the open field test was cleaned with 10% ethanol between test sessions. The first open field tests session was performed at the 13th week of age and was repeated once per week for the subsequent 5 weeks in the ALS group (ALS 1 onset and ALS TER), ALS SWIM group (ALS 1 onset SWIM and ALS TER SWIM), WT group (WT 1 onset and WT TER), and WT SWIM group (WT 1 onset SWIM and WT TER SWIM). The open field test in the SWIM groups was performed on days without training session.

### 4.6. Measurement of Enzymes Activities

All the enzyme activities were measured in the spinal cord homogenates spun in 750 g. All of the samples were analyzed in duplicate. Due to a limited amount of collected tissue, not all samples ware assayed for all of the measurements. The CS activity was measured according to De Lisio et al. [50]. Briefly, the tissue was incubated for 2 min in 970 μL of buffer (50 mM Tris-HCl, 1 mM EDTA, 0.05% Triton-X100, pH 7.8), 10 μL of freshly made DTNB (10 mM) and 10 μL acetyl-CoA (50 mM). The reaction was initiated with the addition of oxaloacetic acid (10 mM). The CS activity was measured at 37 °C, in duplicates. The absorbance was read at 412 nm (Cecil CE9200, Cecil Instruments Limited, Cambridge, UK).

The CAT activity was determined by measuring the kinetic decomposition of H_2_O_2_ according to Aebi [51]. Briefly, 20 μL of homogenate was added to 970 μL of phosphate buffer (50 mM with 5 mM of EDTA, and 0.05% Triton X-100 at pH 7.4). Then, 10 μL of 1 M H_2_O_2_ was added to the quartz cuvette and mixed to initiate the reaction. Absorbance was measured at 240 nm for 1 min at 25°C (Cecil CE9200, Cecil Instruments Limited, Cambridge UK). All of the samples were analyzed in duplicate.

The LDH activity was measured as previously described [52] with a slight modification. Briefly, 6 μL of tissue homogenates were incubated for 2 min in 237 μL of buffer (50 mM potassium phosphate buffer, 0.1 mM EDTA, pH 7.4) and 4.5 μL of NADH (10 mM). The reaction was initiated with the addition of 2.5 μL of pyruvate (210 mM), to a total volume of 250 μL. The LDH activity was measured at 30 °C, in duplicates. The absorbance was read at 340 nm in a microplate reader, Thermo Scientific Multiscan Go (ThermoFisher Scientific, Vartaa, Finland).

The SOD and SOD2 (mitochondrial MnSOD) activities were measured with a superoxide dismutase assay kit (706002, Cayman Chemical, Michigan, MI, USA) according to the manufacturer’s instructions. The SOD1 (cytosolic Cu/Zn-SOD) activity was calculated by subtracting SOD2 activity from the SOD total activity. All of the samples were analyzed in duplicate in a microplate reader, Thermo Scientific Multiscan Go (ThermoFisher Scientific, Vartaa, Finland).

The GPx activity was determined with a Cayman GPx Assay Kit (703102, Cayman Chemical, Michigan, MI, USA) according to the manufacturer’s instructions. All of the samples were analyzed in duplicate in a microplate reader, Thermo Scientific Multiscan Go (ThermoFisher Scientific, Vartaa, Finland).

### 4.7. Measurement of Oxidative Stress Markers

All the concentrations of the oxidative stress markers were measured in the spinal cord homogenates with the addition of BHT, centrifuged at 5000 *g*. All of the samples were analyzed in duplicate. Due to a limited amount of collected tissue, not all samples ware assayed for all of the measurements.

A biomarker of lipid peroxidation, skeletal muscle 8-Isoprostanes content was determined with an 8-isoprostane ELISA Kit (516351, Cayman Chemical, Michigan, MI, USA) according to the manufacturer’s instructions.

The concentration of all three oxidized guanine species: 8-hydroxy-2′-deoxyguanosine from DNA, 8-hydroxyguanosine from RNA, and 8-hydroxyguanine from either DNA or RNA was determined with DNA/RNA Oxidative Damage (High Sensitivity) ELISA Kit (589320, Cayman Chemical, Michigan, MI, USA) according to the manufacturer’s instructions. 

The concentration of GSH was determined with a glutathione assay kit (703002, Cayman Chemical, Michigan, MI, USA) according to the manufacturer’s instructions.

The SH groups were measured as previously described [7]. In short, 20 μL of homogenate was added to 200 μL of buffer (10 mM potassium phosphate buffer, pH 8.0). Then, 30 μL of SDS (10%) and 30 μL of DTNB (1 mM) were added. The probes were incubated for 30 min at 37 °C. The absorbance was read at 412 nm. The values of the SH groups were calculated against a sample blank (without DTNB) from the standard curve of GSH.

### 4.8. Statistical Analysis

All statistical analyses were performed using the Statistica software program (Statistica 13.1). The results are expressed as a mean ± standard error. The normality of data was tested using the Shapiro-Wilks W-test. The level of significance was set at 0.05 for all analyses. The differences between groups were tested using a one-way ANOVA followed by the least significant difference (LSD) *post hoc* test. The differences in repeated open field tests were analyzed using repeated measures of variance (ANOVA), followed by the LSD *post hoc* test. Associations among measured parameters were analyzed using Pearson’s linear regression (coefficient, r). *p* values less than 0.05 were considered statistically significant.

## 5. Conclusions

The positive impact of swim training might be an effect of both the adaptation to mild oxidative stress of physical exercise and the behavioral stress release. Here we show that swim training significantly reduced hyperactivity of ALS pre-symptomatic mice. Moreover, the adaptation to the swim training includes an increase of glycolytic energy metabolism and soothes the destructive effects of oxidative stress which is associated with a better compensatory antioxidative defense in the spinal cord (Figure 7).

## Figures and Tables

**Figure 1 ijms-22-11614-f001:**
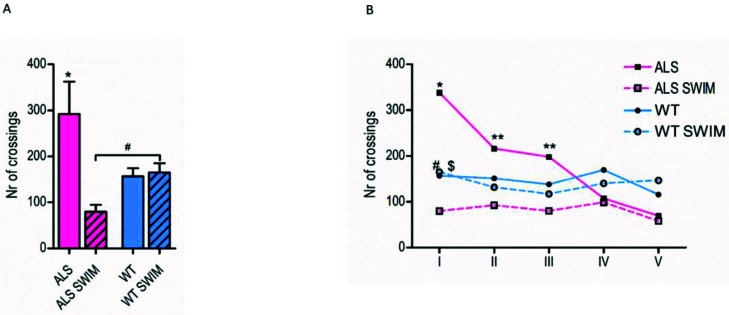
Swim training ameliorates the hyperlocomotor activity during the open field test: (**A**) Number of lines crossed during the first open field test at age 86 days, * *p* < 0.05 versus all of the other groups, # *p* < 0.05, LSD *post hoc* tests after one-way ANOVA. (**B**) Number of lines crossed in five consequent weeks, * *p* < 0.05 versus all of the other groups at the same time point and the same group at the III, IV, and V week, ** *p* < 0.05 versus all of the other groups at the same time point and the same group at IV and V week, for WT group # *p* < 0.05 versus the same group at V week, for WT swim group $ *p* < 0.05 versus the same group at III week, LSD *post hoc* tests after repeated measures ANOVA was used for all the figures.

**Figure 2 ijms-22-11614-f002:**
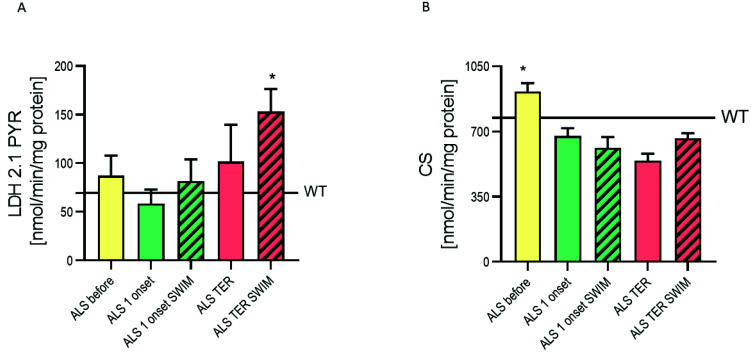
Oxidative and glycolytic enzyme activities in the spinal cord: (**A**) LDH 2.1 PYR activity, WT= 69.5 ± 19.4 nmol/min/mg protein, * *p* < 0.005 versus WT, ALS before and ALS 1 onset groups (WT *n* = 7/8, ALS before *n* = 7/8, ALS 1 onset *n* = 7/8, ALS 1 onset SWIM *n* = 5/7, ALS TER *n*= 5/10, ALS TER SWIM *n* = 9/10). (**B**) CS activity, WT= 773.3 ± 54.1 nmol/min/mg protein, * *p* < 0.05 versus the rest of the ALS groups (WT *n* = 8/8, ALS before *n* = 8/8, ALS 1 onset *n* = 7/8, ALS 1 onset SWIM *n* = 6/7, ALS TER *n*= 7/10, ALS TER SWIM *n* = 10/10). LSD *post hoc* tests after one-way ANOVA was used for all figures.

**Figure 3 ijms-22-11614-f003:**
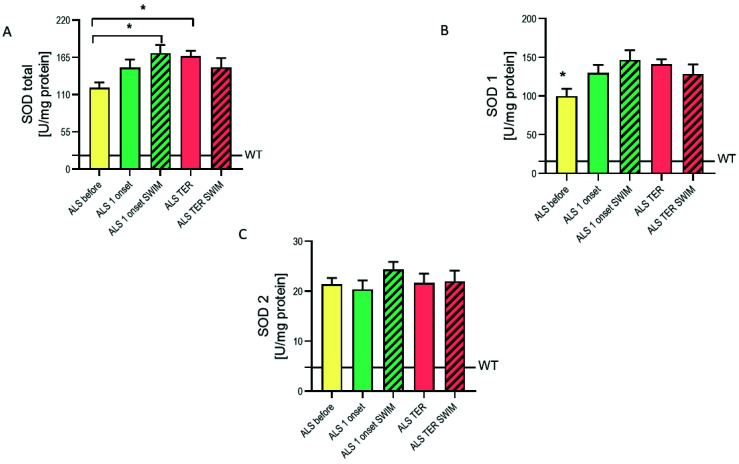
The total superoxide dismutase (SOD), cytosolic SOD and mitochondrial SOD activities in the spinal cord of ALS mice are significantly higher than in TW mice. (**A**) Total SOD activity, WT = 20.34 ± 0.91 U/mg protein, * *p* < 0.05 versus the rest of the ALS groups. (**B**) SOD 1 activity, WT = 15.56 ± 1.3 U/mg protein, * *p* < 0.05 versus ALS 1 onset and ALS TER groups. * *p* < 0.05 with ALS 1 onset and ALS TER groups. For (**A**,**B**) LSD *post hoc* tests were used after one-way ANOVA. (**C**) SOD 2 activity, WT = 4.78 ± 1.05 U/mg protein. For (**A**–**C**): WT *n* = 6/8, ALS before *n* = 8/8, ALS 1 onset *n* = 7/8, ALS 1 onset SWIM *n* = 6/7, ALS TER *n*= 10/10, ALS TER SWIM *n* = 10/10.

**Figure 4 ijms-22-11614-f004:**
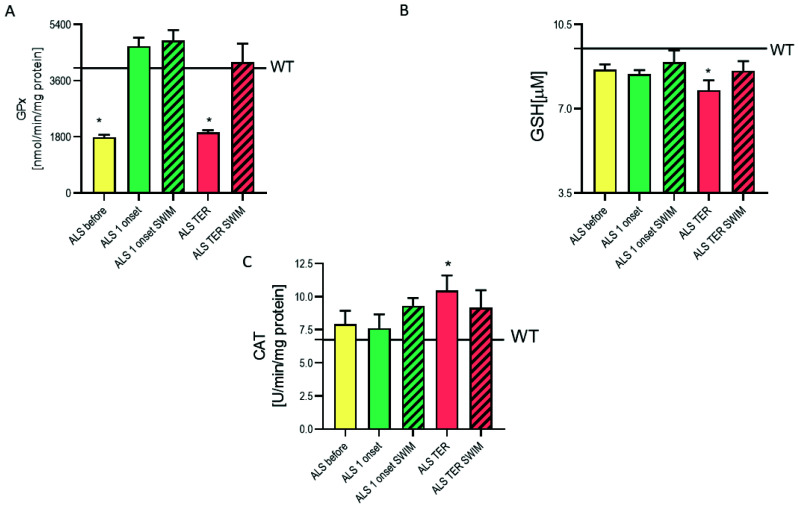
The antioxidative enzyme activities and reduced glutathione (GSH) concentration in the spinal cord: (**A**) Glutathione peroxidase activity, WT = 4001.0 ± 435.4 nmol/min/mg protein, * *p* < 0.05 versus WT and the rest of the ALS groups (WT *n* = 6, ALS before *n* = 8, ALS 1 onset *n* = 7, ALS 1 onset SWIM *n* = 5, ALS TER *n* = 9, ALS TER SWIM *n* = 10). (**B**) GSH concentration, WT = 9.35 ± 0.47 µM, * *p* < 0.05 versus the ALS 1 onset SWIM and WT groups (WT *n* = 7/8, ALS before *n* = 7/8, ALS 1 onset *n* = 7/8, ALS 1 onset SWIM *n* = 6/7, ALS TER *n* = 9/10, ALS TER SWIM *n* = 10/10). (**C**) Catalase activity, WT = 6.7 ± 0.9 U/min/mg protein, * *p* < 0.05 versus the WT group (WT *n* = 8/8, ALS before *n* = 7/8, ALS 1 onset *n* = 6/8, ALS 1 onset SWIM *n* = 5/7, ALS TER *n* = 8/10, ALS TER SWIM *n* = 7/10). For (**A**–**C**) LSD *post hoc* tests were used after one-way ANOVA.

**Figure 5 ijms-22-11614-f005:**
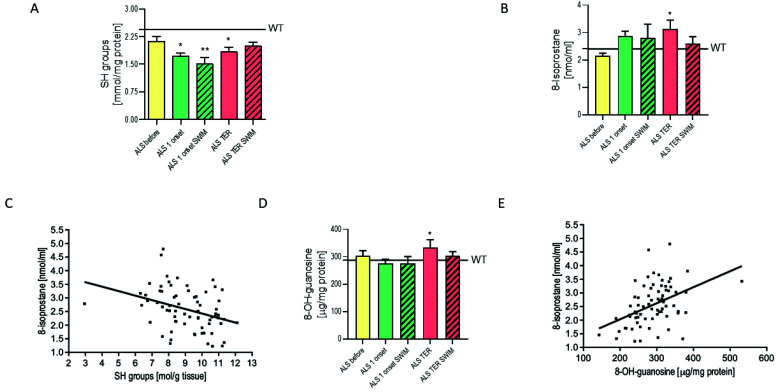
The concentration of the markers of free radical damage in spinal cord: (**A**) The SH groups concentration, WT = 2.43 ± 0.17 mmol/mg protein, * *p* < 0.05 versus the WT, ** *p* < 0.05 versus the ALS before and WT groups (WT *n* = 7, ALS before *n* = 8, ALS 1 onset *n* = 7, ALS 1 onset SWIM *n* = 6, ALS TER *n*= 9, ALS TER SWIM *n* = 10). (**B**) The 8-Isoprostanes concentration, WT = 2.39 ± 0.29 nmol/mL, * *p* < 0.05 versus the ALS before group (WT *n* = 7/8, ALS before *n* = 8/8, ALS 1 onset *n* = 7/8, ALS 1 onset SWIM *n* = 6/7, ALS TER *n*= 9/10, ALS TER SWIM *n* = 10/10). (**C**) The negative correlation of SH groups with 8-Isoprostanes, *p* < 0.05, r = −0.35. (**D**) 8-OH guanosine concentration, WT = 287.2 ± 21.1 µg/mg protein, * *p* < 0.05 versus the ALS 1 onset group (WT *n* = 7/8, ALS before *n* = 8/8, ALS 1 onset *n* = 7/8, ALS 1 onset SWIM *n* = 6/7, ALS TER *n* = 9/10, ALS TER SWIM *n* = 10/10). (**E**) The positive correlation of 8-OH guanosine with 8-Isoprostanes, *p* < 0.05, r = 0.447. For (**A**,**B**,**D**) LSD *post hoc* tests were used after one-way ANOVA.

**Figure 6 ijms-22-11614-f006:**
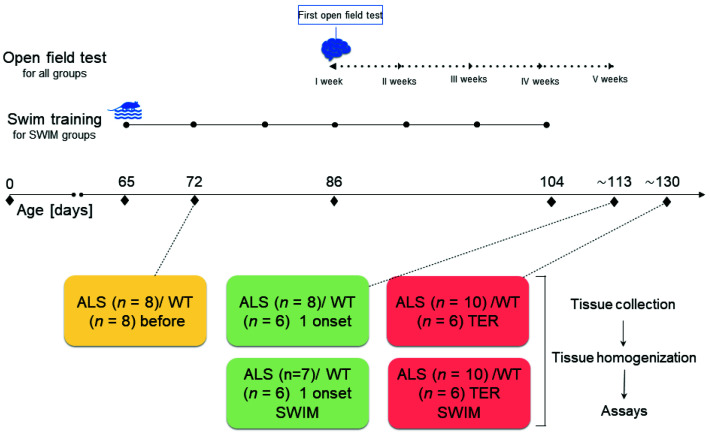
Study design.

**Figure 7 ijms-22-11614-f007:**
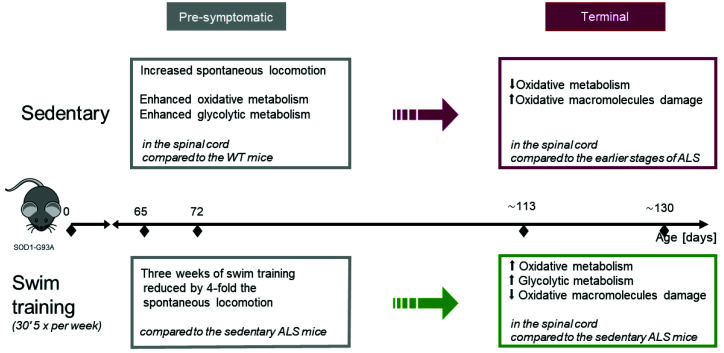
Graphical abstract. ALS disease causes enhanced energy metabolism of the spinal cord. Swim training administrated during the pre-symptomatic phase of ALS causes the reduction of the oxidative stress damage and maintenance of energy metabolism in the spinal cord at the terminal stage of the disease.

## Data Availability

The data sets used and/or analyzed during the current study are available from the corresponding author on the reasonable request.

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
