# Peer review of "Swim Training Ameliorates Hyperlocomotion of ALS Mice and Increases Glutathione Peroxidase Activity in the Spinal Cord"

_ijms, 2021, doi:10.3390/ijms222111614_

Round 1
Reviewer 1 Report
The manuscript entitled "Swim training ameliorates hyperlocomotion of ALS mice and increases gluthatione peroxidase activity in the spinal cord" by Dzik et al. presents a convincing line of evidence that swim training modulates a redox state of motoneurons in ALS mice. The study is planned and performed correctly, the methodology employed is properly described, and the data are analyzed correctly. I have neither meritoric nor methodological concerns, however, I have two questions to the authors:
- The authors did not discussed the observed changes regarding free radical damages - the data are presented only in the Results section. It is documented that GPx has an ability to reduce lipid peroxides, thus, is it possible that decreased 8-isoprostanes level in ALS TER SWIM group is caused by an increased GPx activity in the same mice group?
- The swim training increased CS activity in ALS TER SWIM mice group, thus, it may be concluded that a mitochondrial overal activity also increased within this group. One could suspect that even slightly elevated mitochondrial activity can cause elevated ROS production. May it be a (in- or direct) reason of increased GPx activity in ALS TER SWIM animals?
Maybe it would be interesting to discusse the above issues in the Discussion section?
Please, check the Figure 2. The Fig.2A and Fig.2B citations within the main text do not correspond with the figure's panels. Similarly, the Fig.2 legend does not reflect the panels correctly.
Author Response
Response to the reviewer 1 comments
The manuscript entitled "Swim training ameliorates hyperlocomotion of ALS mice and increases gluthatione peroxidase activity in the spinal cord" by Dzik et al. presents a convincing line of evidence that swim training modulates a redox state of motoneurons in ALS mice. The study is planned and performed correctly, the methodology employed is properly described, and the data are analyzed correctly. I have neither meritoric nor methodological concerns, however, I have two questions to the authors:
- The authors did not discussed the observed changes regarding free radical damages - the data are presented only in the Results section. It is documented that GPx has an ability to reduce lipid peroxides, thus, is it possible that decreased 8-isoprostanes level in ALS TER SWIM group is caused by an increased GPx activity in the same mice group?
Response. Thank you very much for this question. This certainly is a brilliant observation, indeed, the decreased level of 8-isoprostanes definitely might be a result of the defensive role of GPx against lipids peroxidation.
- The swim training increased CS activity in ALS TER SWIM mice group, thus, it may be concluded that a mitochondrial overal activity also increased within this group. One could suspect that even slightly elevated mitochondrial activity can cause elevated ROS production. May it be a (in- or direct) reason of increased GPx activity in ALS TER SWIM animals?
Response. Thank you very much for this question. Of course, this scenario is possible.
Indeed, mitochondria might be at the same time the source and the target of ROS. In our study, we might assume that the pathological changes caused by ALS progression, which manifest themselves with decreased mitochondrial function, were suppressed by the swim training. As the result, we found increased CS activity. We did not measure ROS production/generation, thus, we may only speculate. It is very likely that as a result of swimming training, the ROS altered to the physiological level induces a higher GPx activity. Therefore, elevated GPx activity is a positive adaptation to swimming and plays an essential role against further free-radical damage of macromolecules.
Maybe it would be interesting to discusse the above issues in the Discussion section?
Response: Thank you for this suggestion. That allowed us to see what we have missed. We decided to add a few sentences to the Discussion section.
Please, check the Figure 2. The Fig.2A and Fig.2B citations within the main text do not correspond with the figure's panels. Similarly, the Fig.2 legend does not reflect the panels correctly.
Response: Thank you for these comments. We have made appropriate changes to the text and the figure legend.
Reviewer 2 Report
In the current manuscript, authors studied SOD1-G93A mice and wild-type mice for behavior at different stages of disease progression. They applied Swim training 5 times per week 24 for 30 minutes. All mice underwent behavioral tests. The spinal cord was analyzed for the enzymatic activities and oxidative stress markers. They found pre-symptomatic ALS mice showed increased locomotor activity versus control mice; the swim training reduced these symptoms. The metabolic changes in the spinal cord were present at the pre-symptomatic stage of the disease with the shift towards glycolytic processes at the terminal stage of ALS. Swim training draw the adaptation resulting in a higher glutathione peroxidase and the protection against oxidative stress. Authors conclude that therapeutic aquatic activity might slow down the progression of ALS. These observations are interesting and worth reporting.
Concerns – SOD1 activity is altered with swim training in ALS mice, then what is the mechanistic conclusion, what is data on control mice in Figure 2. It is confusing to see line across indicating WT in most figures – it is important to provide enzymatic activities for WT mice.
Author Response
Response to the reviewer 2 comments
In the current manuscript, authors studied SOD1-G93A mice and wild-type mice for behavior at different stages of disease progression. They applied Swim training 5 times per week 24 for 30 minutes. All mice underwent behavioral tests. The spinal cord was analyzed for the enzymatic activities and oxidative stress markers. They found pre-symptomatic ALS mice showed increased locomotor activity versus control mice; the swim training reduced these symptoms. The metabolic changes in the spinal cord were present at the pre-symptomatic stage of the disease with the shift towards glycolytic processes at the terminal stage of ALS. Swim training draw the adaptation resulting in a higher glutathione peroxidase and the protection against oxidative stress. Authors conclude that therapeutic aquatic activity might slow down the progression of ALS. These observations are interesting and worth reporting.
Concerns – SOD1 activity is altered with swim training in ALS mice, then what is the mechanistic conclusion, what is data on control mice in Figure 2. It is confusing to see line across indicating WT in most figures – it is important to provide enzymatic activities for WT mice.
Response: Thank you for the comment. Indeed, the SOD1 activity in the ALS 1 outset SWIM group is higher than in the rest of the groups, however, this is not a significant change. Nevertheless, we assume, that it might be an adaptational response to the increased ROS production during swim training.
We know that the line for the WT mice might be confusing, however, we already show five columns for ALS groups on most of the graphs and we think that the sixth for the WT group might be too much.
We decided to add the values of WT enzymatic activities to the figure legend of every graph where we used the line instead of the column.